# Small RNAs as biomarkers to differentiate benign and malign prostate diseases: An alternative for transrectal punch biopsy of the prostate?

Lukas Markert[1][☯]*, Jonas Holdmann[1][☯], Claudia Klinger[1,2], Michael Kaufmann[1,2], Karin Schork[3,4], Michael Turewicz[3,4], Martin Eisenacher[3,4], Andreas Savelsbergh[1,2]

1 Division of Functional Genomics, Chair for Biochemistry and Molecular Medicine, Witten/Herdecke University, Witten, Germany, 2 Centre for Biomedical Education and Research (ZBAF), Witten/Herdecke University, Witten, Germany, 3 Medizinisches Proteom-Centre, Ruhr-University Bochum, Bochum, Germany, 4 Centre for Protein Diagnostics (ProDi), Medical Proteome Analysis, Ruhr-University, Bochum, Germany

☯ These authors contributed equally to this work.

* lukas.markert@uni-wh.de

**Data Availability Statement:** The raw data is submitted at biostudies (ebi.ac.uk). The datasets used and analysed during the current study are

## Abstract

Prostate cancer (PCa) is the most common cancer and the third most frequent cause of male cancer death in Germany. MicroRNAs (miRNA) appear to be involved in the development and progression of PCa. A diagnostic differentiation from benign prostate hyperplasia (BPH) is often only possible through transrectal punch biopsy. This procedure is described as painful and carries risks. It was investigated whether urinary miRNAs can be used as biomarkers to differentiate the prostate diseases above. Therefore urine samples from urological patients with BPH (25) or PCa (28) were analysed using Next-Generation Sequencing to detect the expression profile of total and exosomal miRNA/piRNA.

79 miRNAs and 5 piwi-interacting RNAs (piRNAs) were significantly differentially expressed (adjusted p-value < 0.05 and log2-Fc > 1 or < -1). Of these, 6 miRNAs and 2 piRNAs could be statistically validated (AUC on test cohort > = 0.7). In addition, machine-learning algorithms were used to identify a panel of 22 additional miRNAs, whose interaction makes it possible to differentiate the groups as well. There are promising individual candidates for potential use as biomarkers in prostate cancer. The innovative approach of applying machine learning methods to this kind of data could lead to further small RNAs coming into scientific focus, which have so far been neglected.

## Introduction

The prostate is an accessory sex gland. It is located on the underside of the bladder and enclosing the first part of the urethra. Histologically and clinically relevant three concentric zones are distinguished: central/internal zone, transition zone and peripheral/outer zone [1]. The

available at the following data repository: https://www.ebi.ac.uk/biostudies with the number S-BSST586.

**Funding:** LM, JH, AS, CK, MK and AS were funded by the Paul-Kuth-Foundation, Wuppertal, Germany, for this study with 43.175€. (AZ310, https://www.im.nrw/paul-kuth-stiftung) The funding institution did not have any influence on the setting, implementation, analysis, evaluation and publication of the study. The work of KS, MT and ME was supported by de.NBI (FKZ 031 A 534A), a project of the German Federal Ministry of Education and Research. The funding of ME relates to PURE and VALIBIO, projects of Northrhine-Westphalia. (https://www.denbi.de).

**Competing interests:** The authors have declared that no competing interests exist.

prostate produces exocrine about one third of the ejaculate volume and thus serves the survival and motility of the sperm.

With increasing age, the tissue of the transition and central zone surrounding the urethra tends to benign enlargement, benign prostatic hyperplasia (BPH). About 75% of men between the ages of 60 and 69 are affected by BPH [2]. In autopsy studies, the prevalence increases up to 86% of people over 80 years of age [3].

In various stages it can show very similar symptoms as the prostate carcinoma (PCa) e.g. reduction of urinary stream, urinary retention or infections of the urinary tract, yet the outcome is essentially different.

Prostate cancer is the most common cancer and the third most frequent cause of male cancer death in Germany [4] and worldwide [5]. Autopsies revealed that 59% of over 79-year-olds had PCa, probably unnoticed by the patient [6, 7]. In 95% it is an adenocarcinoma, mostly located peripheral.

For the diagnostic differentiation between those two diseases basically digital rectal examination, transrectal ultrasound and PSA-levels in blood are used. More elaborate methods are CT and MRI. Since these methods are quite error-prone and/or resourceful, the gold standard for the diagnosis is the trans-rectal punch biopsy [8]. This procedure is described as painful and carries risks for the patient. A cancer classification is then made according to the staging system TNM (Classification of Malignant Tumours) and Gleason score, which distinguishes the risk groups. For the five degrees of malignancy according to Gleason, the primary growth profile of the prostate cells and the second most common profile are named by a point system [9]. The addition of the two values gives the final score. This results in values from 2 (1+1) to 10 (5+5). This makes it possible to estimate the extent of malignancy within a biopsy.

The aim of this study is to investigate the suitability of small RNA from urine for non-invasive differentiation of dignity of prostate diseases. The group of small RNAs includes seven kinds of non-coding RNAs, that are up to 200 nt long. This project focuses on the established micro-RNAs (miRNA) and the promising piwi-interacting-RNAs (piRNAs). MiRNAs appear to be involved in the development and progression of PCa [10]. MiRNAs are small, non-coding RNAs with a length of 20–25 nucleotides. Parts of the miRNA show complementarity to mRNAs (messenger RNA) that can be influenced in their function by attachment. Different miRNAs on one mRNA but also one miRNA on several mRNAs can act simultaneously [11]. It is estimated that two thirds of all human genes are co-regulated by non-coding miRNAs and that nearly every cellular metabolic process can be influenced [12–14]. In connection with malignancies, the effect on both oncogenes and tumour suppressor genes could be shown [15]. PiRNAs play a role in the silencing of transcription during gametogenesis by this and that mechanism.

It should be pointed that the purpose here is to distinguish between two relatively common diseases, not to test against a healthy cohort. This could be the base for creating an alternative biomarker tool based on miRNA and/or piRNA expression in the diagnostics and possibly avoid punch biopsies. As small RNAs can be easily isolated and urine is non-invasively accessible, we concentrated on this approach.

## Methods and materials

### Sample collection

The study was approved by the ethics committee of the University of Witten/Herdecke by the ethics vote 07/2014. A subsequent registration of the names and project plans of the participants took place in October 2018 without objections from the same ethics committee. The study is subject to the Declaration of Helsinki in its last revised version of 2013. All

experiments were performed in accordance with the relevant guidelines and regulations. Informed consent was obtained written from all participants and/or legal representatives (if they have been nominated by the German guardianship court) for the approval of laboratory work with the patient samples used. The informed consent procedures were reviewed and approved by the Ethics Committee. For the study, 85 patients (between 44 and 86 years old) were recruited who underwent a prostate biopsy due to diagnostic confirmation of prostate enlargement and elevated PSA levels. The samples were collected between 02/2014 and 02/2015 at the Clinic for Urology and Paediatric Urology of the *Helios Klinikum Wuppertal* and immediately stored at -80°C. From the participants, 10 ml urine, 5 ml saliva and 2 x 9 ml blood samples were obtained. The samples were taken during the routine blood collection in the inpatient stay. No separate punctures were necessary. In addition, the core clinical data of their prostate disease were recorded, listed as results in Table 1. Exclusion criteria for study participation were inability to actively consent to the study, infectious diseases (e.g. HIV, hepatitis, TB, etc.), urinary diversion using intestinal segments, saliva-reducing diseases, radiation therapy of the small pelvis or a history of anti-androgenic therapy. Based on the biopsy results, two groups were formed: patients with prostate carcinoma (n = 53) and patients with benign prostatic hyperplasia (n = 32). Of the 85 patient samples, a selection was made according to the integrity of the patient data. Since a comparison of other miRNA extraction methods was carried out at least 8 ml of urine per patient was required for the inclusion of the samples. As a result, 53 patient samples could be selected and processed, 28 with prostate carcinoma and 25 with benign prostate hyperplasia. For the main evaluation, 15 samples (BPH) were compared with 18 samples (PCa) and the results were again validated against the remaining samples (n = 20). Since there have been no previous studies on this concrete issue, we were missing the variances of the individual miRNAs and the expected differences between the two groups. So, we could not make a valid power-calculation in advance and relied on the assessment of the resident statisticians (n>40). The division was done in this way to keep the validation group robust enough. For the random split of samples, two-thirds were set aside for bioinformatical training and one-third for testing.

**Table 1. Clinical data of the study participants.**

| Characteristics | BPH-Patients | PCa-Patients |
|---|---|---|
| **Number (n)** | 25 | 28 |
| **Age (years)** | | |
| Median | 68 | 69,5 |
| Age (range) | 47–86 | 44–84 |
| **PSA** | | |
| Total in ng/ml (median) | 6.89 | 8,34 |
| PSA in ng/ml (range) | 0.52–29.45 | 2.64–228, (outlier: 3193) |
| fPSA in ng/ml (median) | 1.43 | 0.965 |
| PSA-Ratio (median) | 0.18 | 0.16 |
| PSA-Ratio (range) | 0.07–0.46 | 0.07–0.28 |
| **Prostate volume** | | |
| Volume in ml (median) | 50 | 38.5 |
| Volume in ml (range) | 21–240 | 12–95 |
| **Urinary status bact. pos**. | 56% (n = 14) | 14% (n = 4) |

## From urine to small RNA-Library

The samples were defrosted and aliquoted in a water bath at 2˚C. For total RNA-extraction the *Urine microRNA Purification Kit (Cat. 29000*, Norgen Biotek Corp, Thorold, Canada) was used according to the manufacturer's instructions. The concentration of the preparations was measured using *Qubit Fluorimeter™ 3.0* by Invitrogen Life Technologies Corp., Carlsbad, CA, USA.

The library preparation was done with the *QIAseq miRNA Library Kit* (Qiagen, Hilden, Germany) according to the manufacturer instructions from August 2018. An extended 3' ligation of 18 hours was performed to increase efficiency. In all steps industrially purified nuclease-free water was used, which was treated without DEPC. Apart from this, the manufacturer's instructions and specified concentration ranges were followed precisely. During the cDNA synthesis (reverse transcriptase reaction), unique molecular identifiers (UMI) were integrated which mark the original miRNA input quantity and thus counteract a quantitative bias after the amplification reaction.

The libraries were quantified with the *Qubit Fluorimeter™ 3.0* (Invitrogen Life Technologies Corp., Carlsbad, CA, USA) using the Qubit™ microRNA Assay Kit (Cat. Q32881). Quality control was done using polyacrylamide gel electrophoresis (PAGE) and capillary electrophoresis on the *Agilent 2100 Bioanalyzer™* (Kit Agilent DNA 1000, Agilent, Santa Klara, CA, USA). Next Generation Sequencing was performed using Illumina NextSeq 500™. The reading length was 75 base pairs, which also allowed the reading of the attached unique molecular identifiers.

## Bioinformatical and statistical analysis

For raw data analysis, the *data analysis centre* by Qiagen N.V. (qiagen.com) was used. Here the adapters were removed and the original reads were adjusted to real numbers using the UMIs. The resulting sequences were mapped with the databases miRBase version 21 [16, 17], miRBase hairpin [16], mRNAandotherRNA [18] and piRNABank [19]. The downstream analysis and the graphics were conducted using *R version 3.5.3 and 3.6.0* [20] and R scripts provided by the de.NBI service centre BioInfra.Prot located at Ruhr-University Bochum [21].

As a pre-processing step miRNA and piRNAs with a variance near or equal to zero or with many zero counts (more than three in both groups) were removed. This kept 1795 of 2539 miRNAs and 64 of 265 piRNAs within the total small RNA dataset. The following analysis steps were performed separately for the respective miRNA and piRNA datasets. The samples were split randomly into a training dataset (18 PCA + 15 BPH) and a test dataset (10 PCA + 10 BPH for validation). The *DESeq2 R package version 1.22.2* [22] was used to normalise the data and calculate p-values and fold changes for the comparison of PCa and BPH. In addition, the p-values were adjusted (padj) using the Benjamini-Hochberg procedure to correct for multiple testing. Furthermore, the logarithm to base 2 was determined from the fold change values (Log2-Fc).

The statistical analysis with DESeq2 was performed on the training dataset and thresholds of an adjusted p-value $< 0.05$ and a fold change $> 2$ or $< 0.5$ were applied as selection criteria in order to obtain biomarker candidates. Each single biomarker candidate was evaluated on the yet unused test dataset using a receiver operating characteristic analysis (ROC). ROC curves and area under the curve (AUC) values were calculated to assess the performance of the biomarker candidates on new data.

Additionally, biomarker panels consisting of several miRNA that are able to discriminate well between PCA and BPH were searched using a machine learning-(ML-)based classification algorithm. For this analysis, the training and test dataset were normalised together. Variables,

i.e., miRNAs with a too high correlation (Pearson´s correlation coefficient $\geq 0.8$) were removed and only a representative variable was kept (leaving 1035 miRNAs).

Following a random forest [23] model with 100 000 trees was trained on the training dataset using the R-package randomForest [24] in order to obtain a random forest-specific variable importance measure (i.e. mean decrease of the Gini index) for each miRNA. For panel sizes $m$ between 1 and 50, the $m$ variables with the highest variable importance in the model were selected as a candidate panel. Then, the random forest was retrained with only these variables on the training dataset. All 50 specific models (one for each candidate panel of size $m$) were evaluated by classification of the independent test dataset and drawing ROC curves. Finally, the panel with the best AUC was chosen as the final panel. An additional ROC-curve was drawn to show the performance of the chosen panel on the training dataset (by splitting it 1000 times randomly into a training and test subset with the ratio 2:1). The miRNAs from the chosen panel were visualised in a heat map (z-scored values) with dendrograms from hierarchical clustering.

## Results

### Clinical data

Since the recruitment of the patients followed their order of appearance in the clinic, we had a natural randomisation. The clinical data of the groups is shown in Table 1.

Furthermore, the gleason-score of all malignant diseased patients is shown in Table 2.

The tables show that equivalent cohorts were matched, and a diverse range of disease samples was present.

### MiRNA

Normalisation as described above has resulted in more consistent and comparable data (Fig 1). The distribution of miRNA counts was adjusted to a common level.

In the following analysis we found 105 miRNAs significantly deviating between the two groups (padj $< 0.05$). The top candidates are listed in Table 3 while the complete data are shown in the supplementary part. Thereof 74 were under- and 31 overexpressed in the PCa group. We filtered further for a logFc $< -1$ and $>1$ and obtained 79 miRNAs as biomarker candidates. In the following, we will refer to "differentially expressed miRNAs" as those, who fulfil both filter criteria (based on adjusted p-value and logFc). The values in the tables result from the described procedure. In addition, the AUC values of all 79 included candidates were calculated on the test dataset for validation.

Using the Volcano plot (Fig 2), the miRNA expression changes can be evaluated with regard to their statistical significance. Clear deflections are visible in both directions. The plot shows that a large number of miRNAs is differentially expressed. The closer one gets to the upper corners, the more distinct the boundaries of the findings are.

### PiRNA

Additionally, other small RNAs were determined through the extraction procedure used in this study. Due to their comparable size, they were also isolated using the filter-based method.

**Table 2. Gleason-scores of the study participants.**

| Score | Gleason 6 | Gleason 7a | Gleason7b | Gleason 8 | Gleason 9 |
|---|---|---|---|---|---|
| n = 28 (PCa) | 6 | 10 | 9 | 1 | 2 |

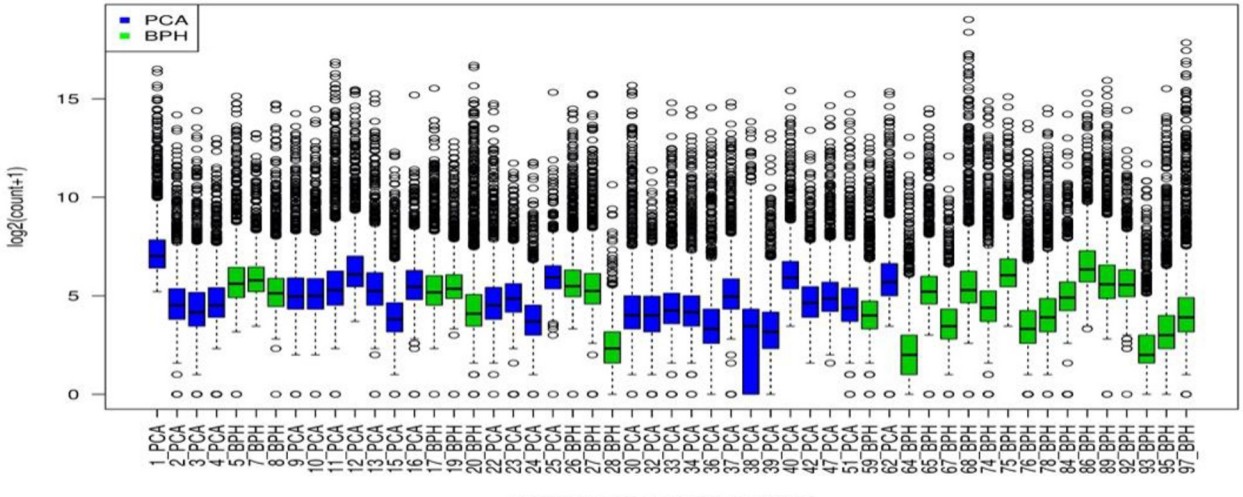

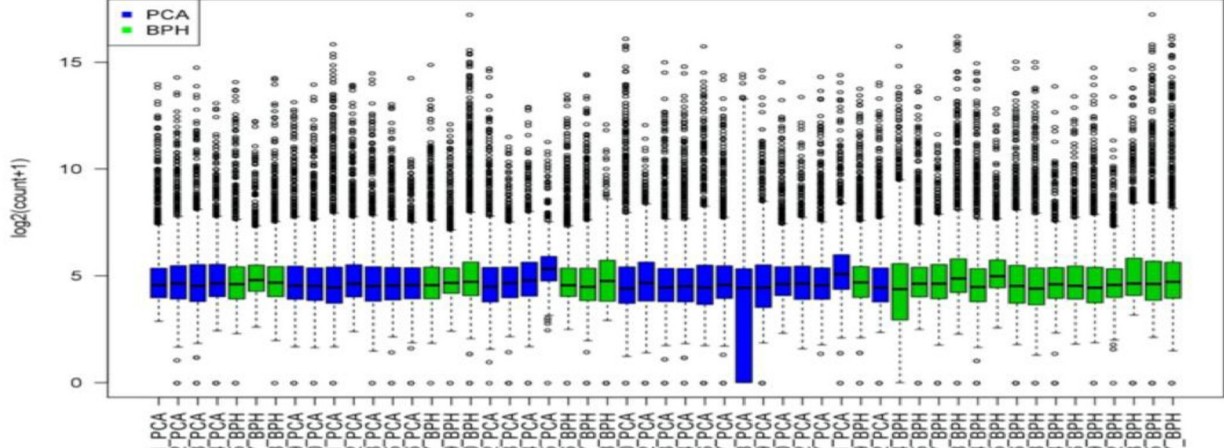

**Fig 1. Distribution of miRNAs/pat.** Before (top) and after (bottom) normalisation.

From the findings we looked at the piRNAs, whose candidates are listed in Table 4. The data were processed and presented as described for miRNAs.

These results are also displayed in a volcano plot (Fig 3). For total piRNAs it shows especially a group of four underexpressed piRNAs in the upper left corner, which stand out as potential candidates for group differentiation. Additionally, a single upregulated piRNA stands out but with less distance to the plot centre.

## Validation

The results of the statistical comparison of the two randomly generated groups were finally validated on the dataset of the remaining samples (test group = 10 BPH vs. 10 PCa). This is shown graphically by ROC curves in Figs 4 and 5. All candidates with an adjusted p-value of $< 0.05$ and a Fold-Change of $> 2$ or $< 0.5$ were examined. The ROC curves show the relationship between false positive rates and true positive rates as well as the area under the curve (AUC). MiRNA or piRNA with AUC values higher or equal than 0.7 was considered as successfully validated. This resulted in six miRNAs (hsa-miR-15p, hsa-miR-3126-3p, hsa-miR-324-5p, hsa-miR-150-5p, hsa-miR-425-3p, hsa-miR-6078), shown in Fig 4 and two piRNAs

**Table 3. Partial representation of differentially expressed miRNAs (complete data available as described).**

| MicroRNA overexpression (PCa vs. BPH) | | | | |
|---|---|---|---|---|
| **miRNA** | **Log2-Fold-Change** | **p-value** | **adjusted p-value** | **AUC(test dataset)** |
| **hsa-miR-6078** | **1.481** | **2.36E-03** | **2.30E-02** | 0.7 |
| hsa-miR-936 | 1.297 | 8.09E-06 | 4.56E-04 | 0.51 |
| hsa-miR-211-5p | 1.295 | 2.04E-04 | 4.60E-03 | 0.67 |
| hsa-miR-6729-3p | 1.294 | 3.95E-04 | 6.91E-03 | 0.44 |
| hsa-miR-891a-5p | 1.275 | 2.01E-03 | 2.09E-02 | 0.64 |
| hsa-miR-6080 | 1.172 | 2.50E-03 | 2.39E-02 | 0.64 |
| hsa-miR-3668 | 1.070 | 4.30E-03 | 3.44E-02 | 0.65 |
| hsa-miR-5088-5p | 1.037 | 7.94E-04 | 1.10E-02 | 0.54 |
| MicroRNA underexpression (PCa vs. BPH) | | | | |
| **miRNA** | **Log2-Fold-Change** | **p-value** | **adjusted p-value** | **AUC(test dataset)** |
| hsa-miR-451a | -4.916 | 1.60E-09 | 9.79E-07 | 0.57 |
| hsa-miR-27a-3p | -3.352 | 1.17E-06 | 1.03E-04 | 0.52 |
| hsa-miR-106b-5p | -3.023 | 2.49E-07 | 3.81E-05 | 0.52 |
| hsa-miR-151b/151a-5p | -2.940 | 4.05E-07 | 4.13E-05 | 0.54 |
| hsa-miR-182-5p | -2.937 | 3.59E-07 | 4.13E-05 | 0.58 |
| hsa-miR-6865-5p | -2.860 | 1.04E-08 | 3.19E-06 | 0.53 |
| hsa-miR-16-2-3p | -2.565 | 7.09E-05 | 2.55E-03 | 0.51 |
| hsa-miR-17-5p | -2.385 | 6.13E-04 | 9.16E-03 | 0.53 |
| hsa-miR-200c-3p | -2.369 | 1.27E-04 | 3.71E-03 | 0.54 |
| hsa-miR-200b-3p | -2.334 | 3.21E-04 | 5.98E-03 | 0.54 |
| hsa-miR-142-5p | -2.228 | 5.87E-03 | 3.99E-02 | 0.46 |
| hsa-miR-20a-5p | -2.177 | 3.23E-04 | 5.98E-03 | 0.52 |
| **hsa-miR-15a-5p** | **-2.175** | **1.29E-03** | **1.51E-02** | **0.71** |
| hsa-miR-320a | -2.163 | 1.76E-04 | 4.49E-03 | 0.59 |
| hsa-miR-146b-5p | -2.092 | 2.23E-04 | 4.70E-03 | 0.55 |
| **hsa-miR-3126-3p** | **-2.057** | **1.12E-07** | **2.28E-05** | **0.76** |
| hsa-miR-126-3p | -2.002 | 1.95E-04 | 4.60E-03 | 0.55 |
| hsa-miR-149-5p | -2.000 | 8.20E-06 | 4.56E-04 | 0.48 |
| **hsa-miR-324-5p** | **-1.404** | **7.60E-04** | **1.08E-02** | **0.74** |
| **hsa-miR-150-5p** | **-1.378** | **2.84E-03** | **2.59E-02** | **0.76** |
| **hsa-miR-425-3p** | **-1.103** | **8.91E-04** | **1.18E-02** | **0.71** |

Overregulations are shown in the upper part and underexpressions in the lower part. All candidates shown correspond to an adjusted p-value <0.05 and a Log2 fold change of <-1 or >1. Bold marked are the miRNAs which were successfully validated on the test dataset (AUC > = 0.7).

(hsa_piR_018849, has_piR_019324) exceeding the set target of at least 0.7 area under the curve (AUC), see Fig 5.

## Panel calculation using machine learning

The panel search was carried out as described above. The panel with the best test set classification (based on AUC values) included the following 22 miRNAs:

hsa-miR-1231, hsa-miR-4726-5p, hsa-miR-6814-3p, hsa-miR-4758-5p, hsa-miR-154-5p,

hsa-miR-7855-5p, hsa-miR-6717-5p, hsa-miR-31388, hsa-miR-181d-3p, hsa-miR-1304-3p,

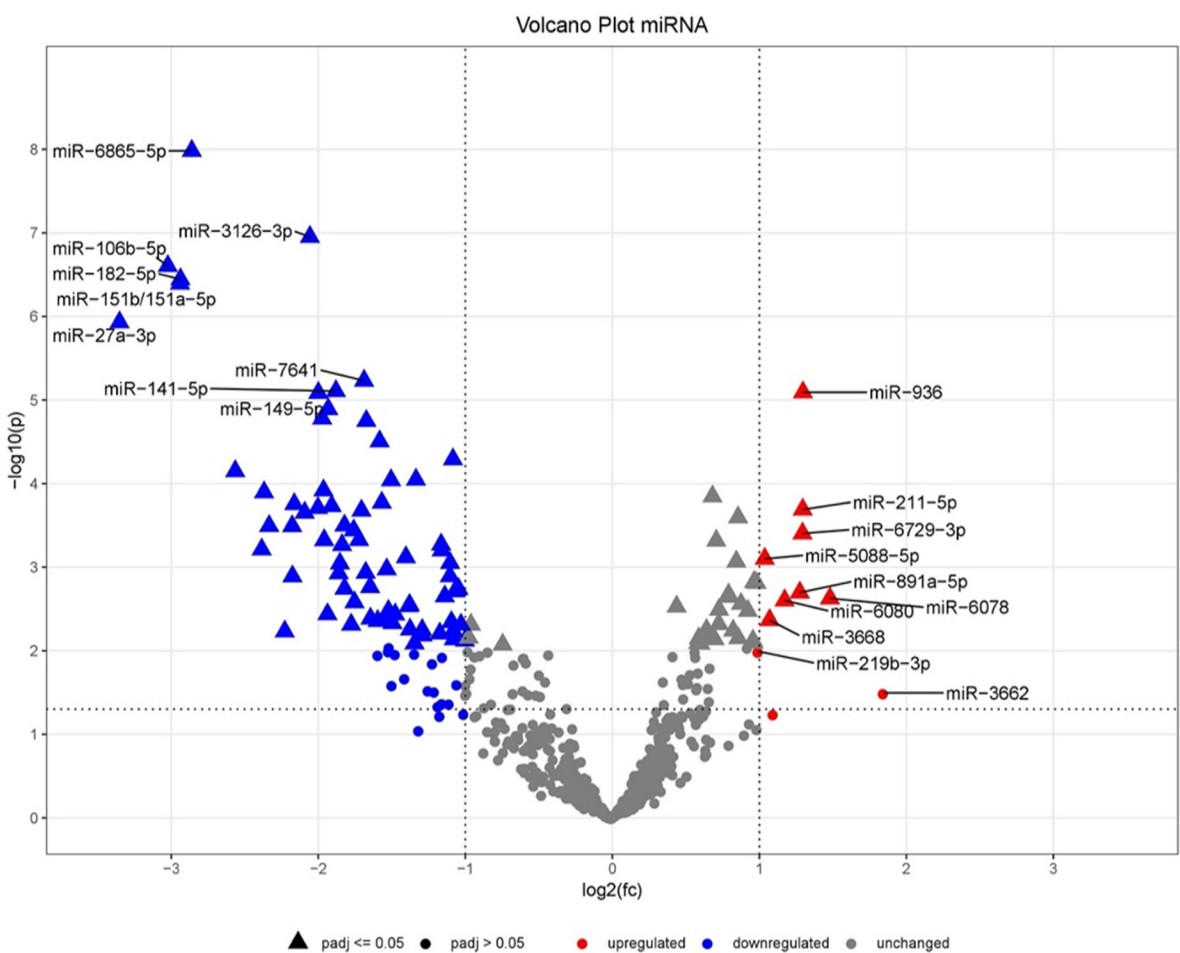

**Fig 2. Volcano plot of the comparison of the miRNAs.**

hsa-miR-1972, hsa-miR-874-3p, hsa-miR-5194, hsa-miR-3689a-5p, hsa-miR-7107-3p, hsa-miR-6878-3p, hsa-miR-5196-5p, hsa-miR-4712-5p, hsa-miR-6515-5p, hsa-miR-6727-3p,

hsa-miR-7111-3p, hsa-miR-1181.

This panel found on the training data record was validated on the test data record with an AUC of 0.7. In a heat map, based on the training dataset, the miRNA panel is visualised (Fig 6).

**Table 4. Differentially expressed total piRNAs.**

| piRNA overexpression (PCa vs. BPH) | | | | |
|---|---|---|---|---|
| **piRNa** | **Log2-Fold-Change** | **p-value** | **adjusted p-value** | **AUC (test dataset)** |
| hsa_piR_020401 | 1.259 | 2.81E-03 | 3.59E-02 | 0.64 |
| piRNA underexpression (PCa vs. BPH) | | | | |
| **piRNa** | **Log2-Fold-Change** | **p-value** | **adjusted p-value** | **AUC (test dataset)** |
| hsa_piR_001318 | -2.110 | 1.25E-05 | 5.32E-04 | 0.56 |
| hsa_piR_018573 | -1.945 | 2.05E-05 | 5.32E-04 | 0.61 |
| **hsa_piR_018849** | **-2.292** | **2.49E-05** | **5.32E-04** | **0.88** |
| **hsa_piR_019324** | **-2.032** | **4.68E-05** | **7.49E-04** | **0.72** |

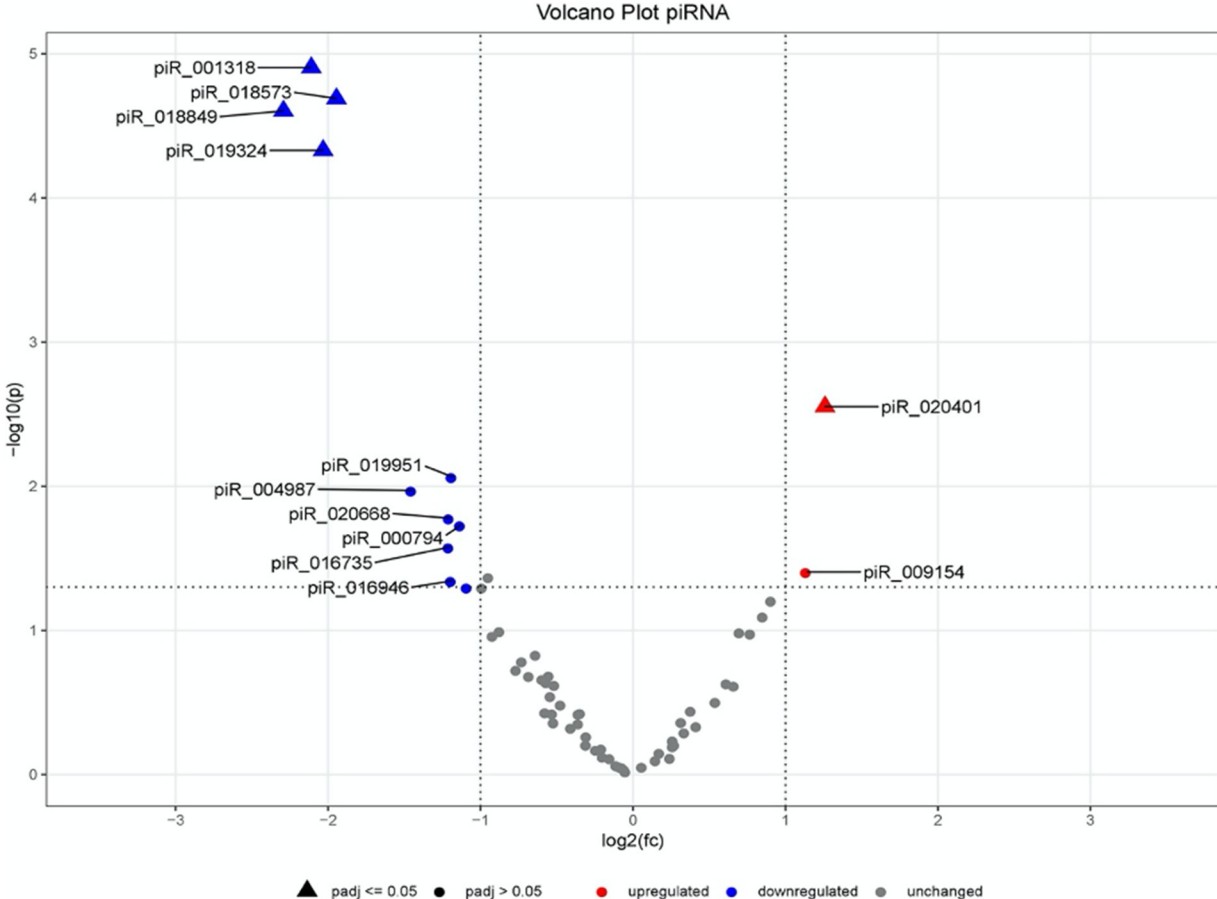

**Fig 3. Volcano plot of the comparison of the piRNAs of PCa vs. BPH.**

Dendrograms stemming from hierarchical clustering were added to the plot to show similarities between miRNAs as well as samples. Dark and light grey colour bars were added to distinguish the two experimental groups. This makes it obvious that the two experimental groups can be distinguished well by the pattern. Rather yellow (top right and bottom left) and rather blue (top left and bottom right) areas of the heatmap show that the panel is a mixture of miRNAs that are higher or lower in the PCa group compared to BPH.

The validation results on the training set are very distinct with an AUC of almost 90%. On the test set the AUC decreases as expected, but still remains at 70% (Fig 7).

## Discussion

Commonly, studies compare healthy and diseased patients. In the present study, patients suffering from two different diseases are compared.

The clinical data of the study participants show that they are comparable groups, as can be seen, for example, at the very similar age distribution. The PSA values deviate from a similar median value in both groups with high variances. The greater organ enlargement of the BPH group in ultrasound matches the higher rate of urinary tract infections clinically due to obstruction. From the Gleason scores, an even distribution of the disease stages of the study participants becomes obvious, indicating the inclusion of a wide variety of PCA disease stages

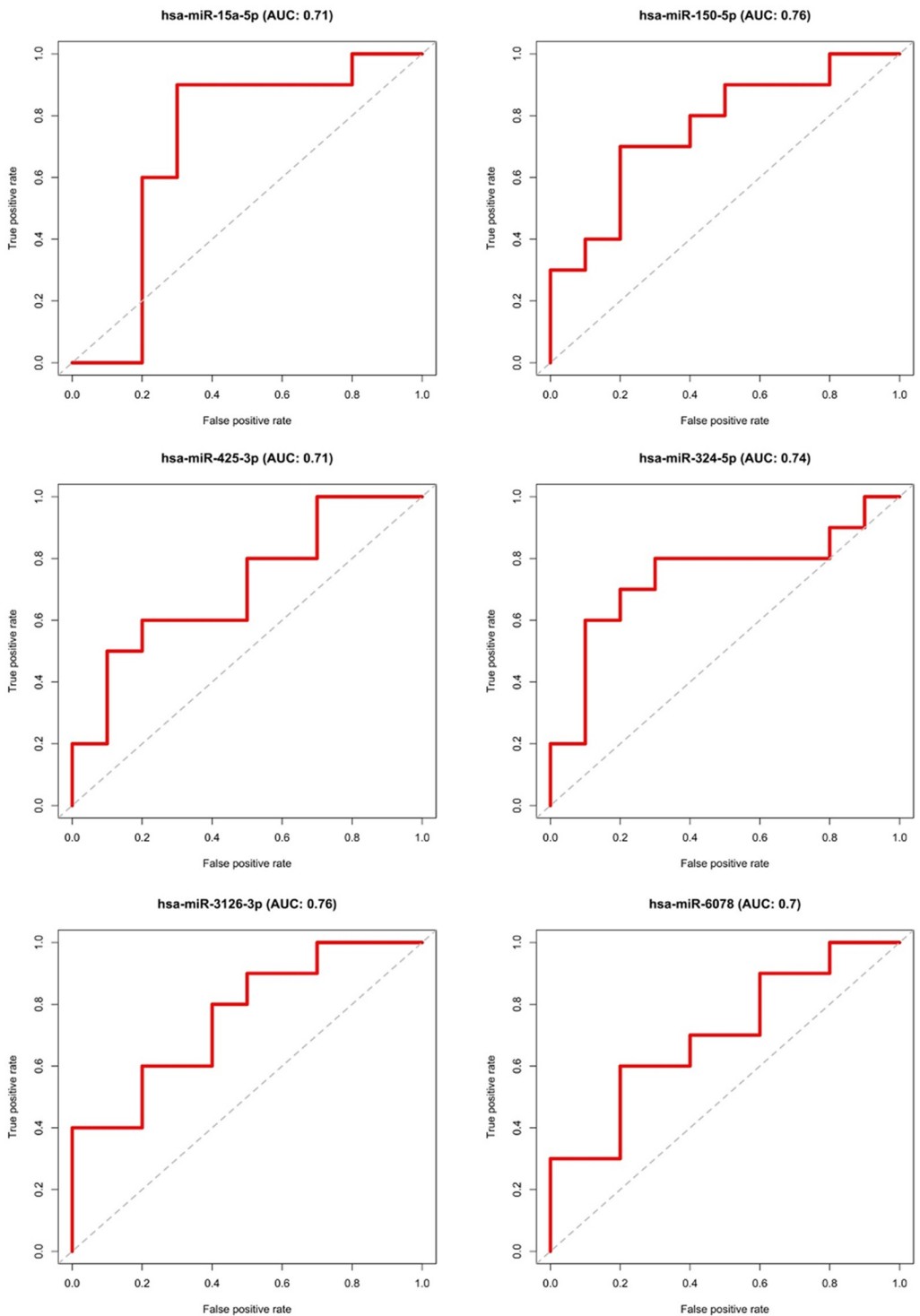

**Fig 4. ROC-curves of the validated miRNAs.**

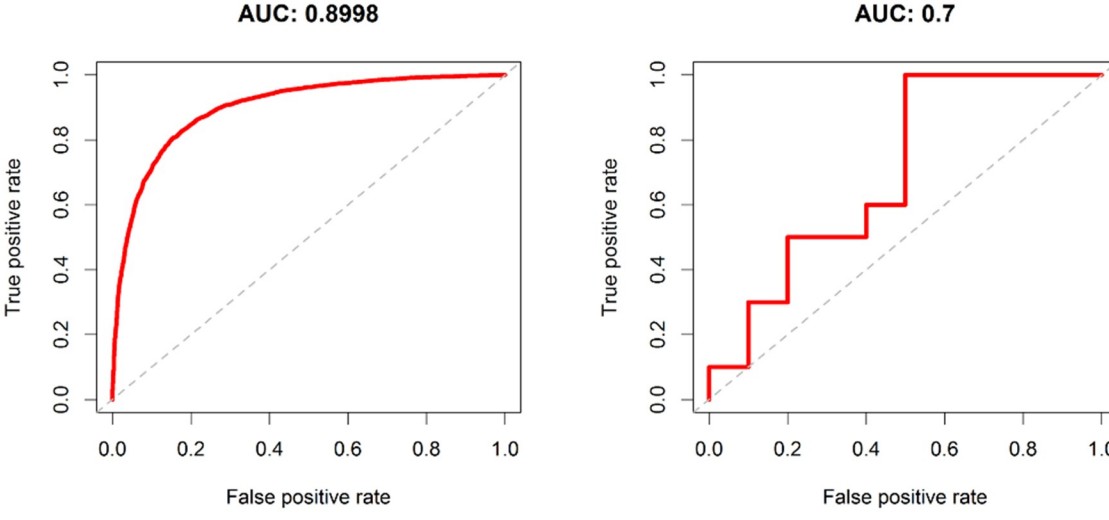

**Fig 5. ROC-curves of the validated piRNAs.**

in the study. Neither only early diagnoses nor only patients with a very unfavourable prognosis (>7 = WHO grade3) were included.

Changes in miRNA expression and activity have been frequently observed and described in PCa and affect PCa initiation and progression in different ways [25]. The following explanations refer to total miRNAs, which were found in our work. It is described that the overexpression of the nuclear MYC (Myelocytomatosis gene) protein has an effect on prostate carcinogenesis [26]. Several miRNAs have been shown to regulate MYC expression in PCa cells, including miR-34, miR-98 and let-7c, either directly or indirectly [27]. MiR-98 and representatives of the let-7 family were also discovered in our study. In other cancer models miR-145 has also been shown to regulate MYC. This gene also directly regulates the transcription of the miR-17-5p cluster, which is frequently deregulated in PCa [28]. Genetic mutation and loss of PTEN (Phosphatase and tensin homolog enzyme) is also a common event in PCa, disrupting the PI3K (phosphoinositide-3-kinase) signalling cascade [29]. MiR-23b, miR-19b, miR-26a and miR-92a have been shown to directly regulate PTEN in PCa cells [30]. In the total miRNA we investigated, miR-19b and miR-26a were significantly downregulated in PCa compared to BPH patients. Another important pathway that is deregulated in advanced metastatic and castration-resistant PCa is the anti-apoptotic protein BCl-2 (B-cell lymphoma *2*) which is an established direct target of miR-15/16, both are significantly downregulated in our samples, miR-15a additionally validated [31]. In addition, several miRNAs were identified as direct regulators of AR expression or AR activity in PCa-progression. These include miR-135b, which was also found in our samples [32]. Abnormal miRNA expression and activity can often be observed and described in PCa and can affect PCa initiation and progression in different ways. This is very clearly summarized by Kumar and Lupold [25]. In addition to those mentioned above, other pathways for PCa initiation and progression are described there (ETS gene fusion, CDKN1B, GSTP1). However, many of these results are based on miRNA expression analyses of tissue samples, which can probably only be compared to those of miRNAs from urine to a limited extent.

In addition, the literature on our found biomarker candidates was examined for effects on other cancers. Interactions in various cancer cells have also been described for the candidates we have validated (hsa-miR-15a-5p (multiple tumour suppressor, among others in chronic

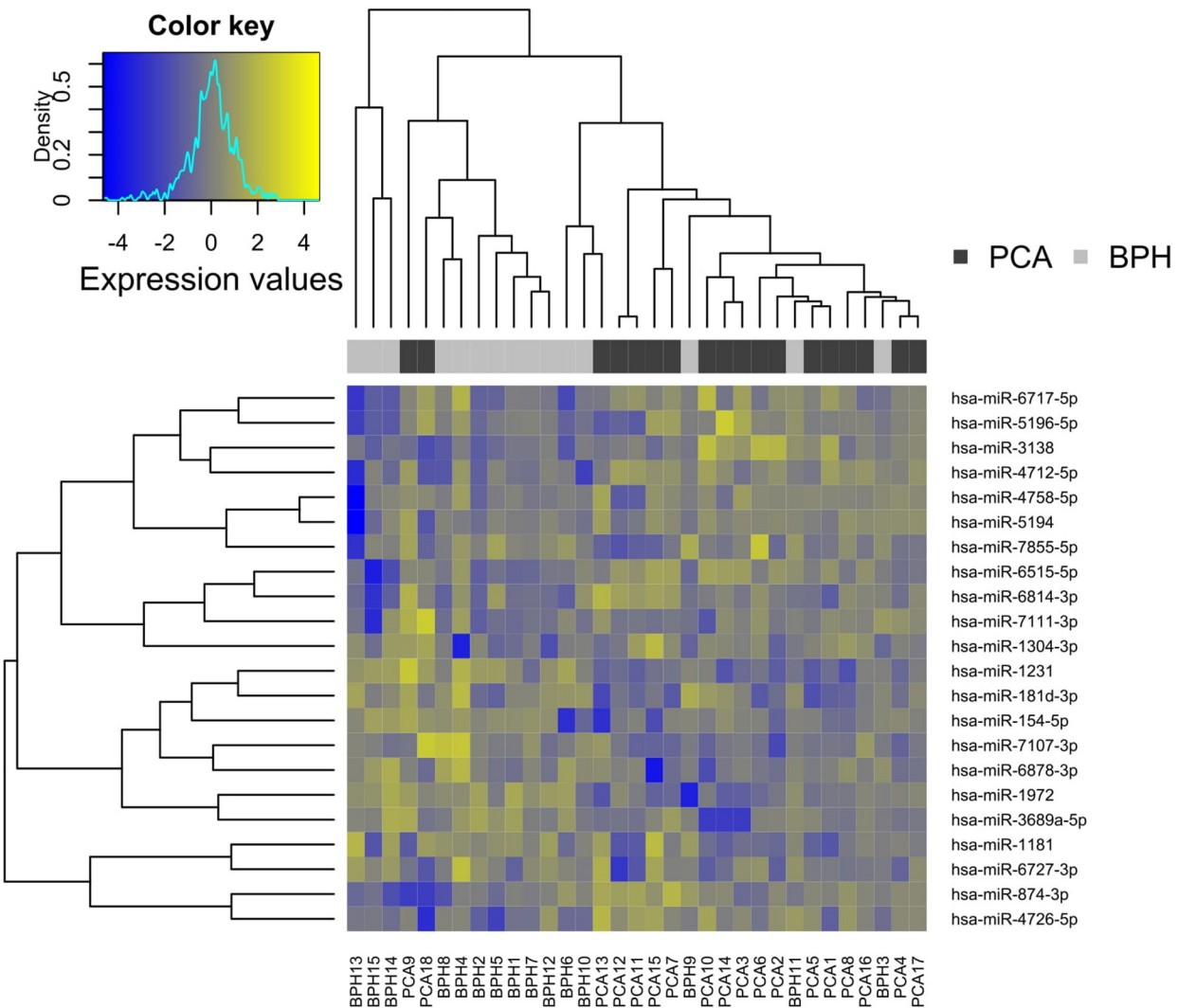

**Fig 6. Heatmap for the identified panel of total miRNAs.**

lymphocytic leukemia [33], bile duct carcinoma [34], pancreas carcinoma [35], prostate carcinoma [36]), hsa-miR-324-5p (downregulated in Glioma [37], Colorectal Carcinoma [38], in hypoxic prostate carcinoma cells [39]) hsa-miR-150-5p (downregulated in acute myeloid leukemia [40], chronic myeloid leukemia [40], Colorectal Carcinoma [41]; upregulated in breast cancer [42], stomach cancer [43], osteosarcoma [44], lung cancer metastasis [45]), hsa-miR-425-3p (overexpressed in stomach carcinoma cells and serum [46], hsa-miR-let-7-5p (often described as tumour suppressor [47, 48]). In most cases, the downregulation in our samples seems to correspond to the described tumour suppressive effect of the mentioned miRNAs.

The correlations found do not allow simple conclusions to be drawn. In different body cells, the same miRNA can apparently act in a disease-specific way to promote or inhibit cancer. It is therefore obviously essential to look at the individual tumour diseases. Moreover, in many cases, cause and effect cannot be clearly distinguished from each other. Thus, the question occasionally arises as to whether over- or under-expression is the trigger or consequence

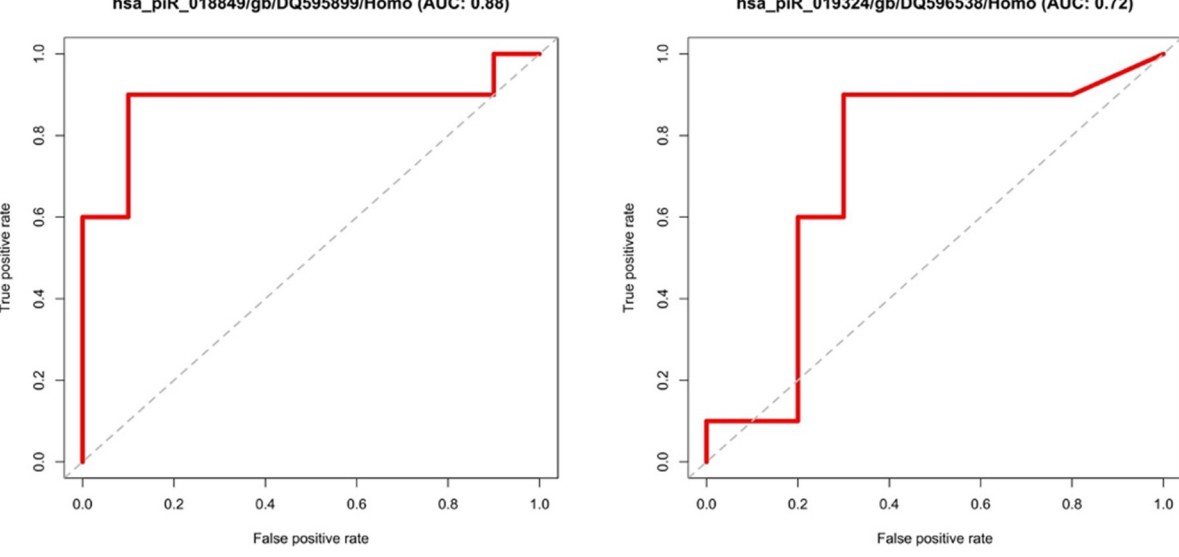

**Fig 7. Validation of the panel of total miRNAs.**

of cell degeneration. For example, DNA methylation can influence miRNA expression, but vice versa, such a deregulation can also influence DNA methylation [25].

For the piRNAs examined here, the validation confirmed the following two candidates: hsa_piR_018849, hsa_piR_019324. They were compared with entries of the piRBase database [49]. For piR_019324 no studies on function and interaction were found. For piR_018849, 95 reference sequences could be specified, but no target gene. Therefore, their molecular function still seems to be in need of clarification. For diagnostic studies, however, they could already be noted. The results of piR_018849 show a log2-fold-change value of -2.292 and robust significance and were validated with a very precise 88% specificity and sensitivity for the best cut-off in the corresponding ROC curve.

After a panel search via machine learning, a heat map was used to display the grouping into clusters. The groups are well distinguishable by the chosen method. The validity of the test data set could directly fulfil the self-imposed quality requirements of an AUC of 0.7. The panel classification based on a set of different biomarkers can be seen as a new promising approach in the diagnosis of prostate carcinoma with miRNAs. The fact that completely different markers are found as individual or panel candidates may be irritating. However, both search methods are totally different, as described. This means that miRNAs appear in the panel that would otherwise not be noticed individually. Should some of these candidates turn out to be highly amplification-stable or particularly pathway-relevant, a panel containing them and providing a disease-specific expression pattern could be highly relevant for further research. Furthermore, such a pattern is probably less susceptible to influences on the expression and analysis of individual markers. Consequently, a diagnostic test based on it may be more reliable.

The panel can only be seen as a complete set—the sum of its individual markers. This results in an innovative inclusion of all generated data and minimizes the dependence on the (error-free) measurability of a single candidate. This approach should be tested on further collectives.

One limitation of this work may certainly be the relatively small sample size. This is due, among other things, to the inclusion and exclusion criteria, the sometimes inadequate amount of sample material and the currently still high costs of next-generation sequencing.

Nevertheless, very convincing results with convincing significance were seen for the comparison of the two main groups. Unfortunately, the distribution of the samples among the different Gleason scores was so heterogeneous that no statistical assignment of the candidates to a specific Gleason score could be made. This seems to be an interesting follow-up point to our work.

## Conclusion

With this study we demonstrated for the first time the possibility to distinguish between benign and malign prostate diseases via small RNA. There are promising individual candidates for potential use as biomarkers in prostate cancer. Moreover, piRNAs also appear to provide different expression profiles in benign and malignant situations. Although direct molecular effects are still mostly unknown, they could be used for diagnostic purposes without additional laboratory expenses. A follow-up of the found markers in the context of larger investigations, at best across several cohorts and research groups, may validate this in the future. It would be important to unify or standardize the evaluation procedure, since the current biochemical and bioinformatical methods of different groups differ from each other and are often based on self-written algorithms with insufficient comparability. The innovative approach of using machine learning algorithms for panel search could lead to further miRNAs, which have so far been neglected, coming into scientific focus. If time and economic efficiency of sequencing continue to improve as it has in recent years, we consider a clinical application of a urine test to be very promising.

## Acknowledgments

Thanks to Fatemeh Gholamrezaei for teaching all practical skills. Many thanks also to Prof. Dr. Florian Kreppel and his colleagues for their kind reception at the chair. We would also like to thank Prof. Dr. Jan Postberg and Dr. Patrick Weil for their helpful and uncomplicated manner in the laboratory in Wuppertal.

## Author Contributions

**Conceptualization:** Lukas Markert, Claudia Klinger, Michael Kaufmann, Andreas Savelsbergh.

**Data curation:** Lukas Markert, Jonas Holdmann.

**Formal analysis:** Lukas Markert, Jonas Holdmann.

**Funding acquisition:** Lukas Markert.

**Investigation:** Lukas Markert, Jonas Holdmann.

**Methodology:** Claudia Klinger, Andreas Savelsbergh.

**Software:** Lukas Markert, Karin Schork, Michael Turewicz.

**Supervision:** Andreas Savelsbergh.

**Validation:** Lukas Markert, Karin Schork, Michael Turewicz.

**Visualization:** Lukas Markert, Karin Schork, Michael Turewicz.

**Writing – original draft:** Lukas Markert, Jonas Holdmann.

**Writing – review & editing:** Lukas Markert, Jonas Holdmann, Claudia Klinger, Karin Schork, Michael Turewicz, Martin Eisenacher, Andreas Savelsbergh.

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
