## [Decision Letter · Decision Letter 0]

21 Jan 2021

PONE-D-20-39305

Small RNAs as biomarkers to differentiate benign and malign prostate diseases: an alternative to transrectal punch biopsy of the prostate?

PLOS ONE

Dear Dr. Markert,

Thank you for submitting your manuscript to PLOS ONE. Your article has been reviewed by an intern Academic Editor and an extern reviewer. After careful consideration, we feel that it has merit but does not fully meet PLOS ONE’s publication criteria as it currently stands. Therefore, we invite you to submit a revised version of the manuscript that addresses the points raised during the review process and indicated below. Please respond adequately to all points.

Furthermore, could you also indicate whether some statistical links have been pointed out between the various identified miRNA and/or piRNA and the Gleason score?

We look forward to receiving your revised manuscript.

Kind regards,

Jean-Marc A Lobaccaro, PhD

Academic Editor

PLOS ONE

Journal Requirements:

2. In your Methods section, please provide additional information about the participant recruitment method and the demographic details of your participants. Please ensure you have provided sufficient details to replicate the analyses such as: a) the recruitment date range (month and year), b) a description of how participants were recruited.

3. Please provide a sample size and power calculation in the Methods, or discuss the reasons for not performing one before study initiation.

4. Please ensure you have discussed any potential limitations of your study in the Discussion, including study design, sample size and/or potential confounders.

5.We note that you state that consent was obtained from legal guardians in some cases. Please explain why patient consent was not appropriate in some cases. Please describe in your methods section how capacity to consent was determined for the participants in this study. Please also state whether your ethics committee approved this consent procedure. If you did not assess capacity please briefly outline why this was not necessary in this case.

6. Please provide additional details regarding participant consent. In the ethics statement in the Methods and online submission information, please ensure that you have specified what type you obtained (for instance, written or verbal, and if verbal, how it was documented and witnessed).

7. We note that you have indicated that data from this study are available upon request. PLOS only allows data to be available upon request if there are legal or ethical restrictions on sharing data publicly. For information on unacceptable data access restrictions, please see http://journals.plos.org/plosone/s/data-availability#loc-unacceptable-data-access-restrictions.

8. Please amend either the title on the online submission form (via Edit Submission) or the title in the manuscript so that they are identical.

9. Your ethics statement should only appear in the Methods section of your manuscript. If your ethics statement is written in any section besides the Methods, please delete it from any other section.

10. We note you have included a table to which you do not refer in the text of your manuscript. Please ensure that you refer to Table 4 in your text; if accepted, production will need this reference to link the reader to the Table.

11. Please include a copy of Table 5 which you refer to in your text on page 12.

Reviewers' comments:

Reviewer's Responses to Questions

**Comments to the Author**

1. Is the manuscript technically sound, and do the data support the conclusions?

Reviewer #1: Partly

2. Has the statistical analysis been performed appropriately and rigorously? 

Reviewer #1: Yes

3. Have the authors made all data underlying the findings in their manuscript fully available?

Reviewer #1: Yes

4. Is the manuscript presented in an intelligible fashion and written in standard English?

Reviewer #1: Yes

5. Review Comments to the Author

Reviewer #1: In this paper, authors investigate wether urinary miRNAs and piRNAs can be used as biomarkers to differentiate benign and malign prostate diseases. This should be an alternative to transrectal punch biopsy of the prostate which is a painful and risky procedure.

For that, urine samples from urological patients with BPH (25) or PCa (28) were analysed using Next-Generation Sequencing to detect the expression profile of total and exosomal miRNA/piRNA. Then a differential study was made and 6 miRNAs and 2 piRNAS were statistically validated

In addition, machine-learning algorithms were used to identify a panel of 22 additional miRNAs.

This paper is of interest and well written in its entirety.

The differential study is clear and robust and conclude to a set of potential biomarkers.

However, the machine-learning part is less clear and its interest is not an evidence for me.

MAJOR remarks and questions:

1) In the paper, samples were split randomly into training dataset (18 PCA + 15 BPH) and a test dataset (10 PCA + 10 BPH) for validation. Sizes of dataset were not justified and it is difficult to know the relevance of this choice. It would be necessary to justify this choice.

2) The interest of additionnal biomaker panels found with machine learning method is not evidence for me. This should be clarified and the interest of such a method be clearly shown.

MINOR remarks:

lines 115-116: maybe only use miRNA and not microRNA

line 166: the reading depth -> maybe reading length

line 166: sentence is not really clear

Figure 2: not centered and truncated

Figure 6: problem of legend for PCA and PH

6. PLOS authors have the option to publish the peer review history of their article (what does this mean?). If published, this will include your full peer review and any attached files.

Reviewer #1: No

---

## [Author Response · Author response to Decision Letter 0]

11 Feb 2021

Response to Reviewers

Dear Sir or Madam!

Thank you very much for your most kind reply with all the helpful questions and comments on our paper. I thank you for the opportunity to respond and would be pleased if the answers are satisfactory to you.

In the following, I will address the individual points, highlighted in color. In addition, you will receive both the marked and the unmarked version of the reviewed paper.

In response to your mail received on January 21st 2021:

The layout was changed to fit the specifications. Pardon for the previous inconsistencies.

2. In your Methods section, please provide additional information about the participant recruitment method and the demographic details of your participants. Please ensure you have provided sufficient details to replicate the analyses such as: a) the recruitment date range (month and year), b) a description of how participants were recruited.

The information was specified in the text. (line 147-174)

3. Please provide a sample size and power calculation in the Methods, or discuss the reasons for not performing one before study initiation.

Due to the absence of preliminary studies on this topic, only a qualified estimate could be made. This has now been explained in the text.

4. Please ensure you have discussed any potential limitations of your study in the Discussion, including study design, sample size and/or potential confounders.

The limitations of the study were named in more detail. (line 479-486)

5.We note that you state that consent was obtained from legal guardians in some cases. Please explain why patient consent was not appropriate in some cases. Please describe in your methods section how capacity to consent was determined for the participants in this study. Please also state whether your ethics committee approved this consent procedure. If you did not assess capacity please briefly outline why this was not necessary in this case.

According to German guardianship law, legally appointed guardians make decisions for their personally known clients in certain medical cases. The prerequisite for the appointment of a guardian is that the person concerned is of adult age and in need of assistance. A person is in need of assistance if, as a result of illness or disability, he or she is no longer able to manage his or her affairs in whole or in part. Of course, in this case, client and legal guardian were asked for written consent.

6. Please provide additional details regarding participant consent. In the ethics statement in the Methods and online submission information, please ensure that you have specified what type you obtained (for instance, written or verbal, and if verbal, how it was documented and witnessed).

Written consent was always obtained. This has now been specified in the text. (line 144)

7. We note that you have indicated that data from this study are available upon request. PLOS only allows data to be available upon request if there are legal or ethical restrictions on sharing data publicly. For information on unacceptable data access restrictions, please see http://journals.plos.org/plosone/s/data-availability#loc-unacceptable-data-access-restrictions.

The relevant raw data will be gladly made available in its entirety. It is submitted at biostudies (https://www.ebi.ac.uk/biostudies) with the number S-BSST586.

There are no objections to the disclosure of the study data. Clear names are not mentioned in it. Regarding all data questions, the Ethics Committee of the University of Witten/Herdecke can be contacted via sekretariat-ethik@uni-wh.de.

We uploaded the complete relevant data at: biostudies (https://www.ebi.ac.uk/biostudies) with the number S-BSST586

8. Please amend either the title on the online submission form (via Edit Submission) or the title in the manuscript so that they are identical.

The title was adapted in the manuscript.

9. Your ethics statement should only appear in the Methods section of your manuscript. If your ethics statement is written in any section besides the Methods, please delete it from any other section.

The text has been adjusted accordingly.

10. We note you have included a table to which you do not refer in the text of your manuscript. Please ensure that you refer to Table 4 in your text; if accepted, production will need this reference to link the reader to the Table.

The numeration mistake has been corrected. (line 139-141)

11. Please include a copy of Table 5 which you refer to in your text on page 12.

 Meant was Table 4, as has now been correctly adjusted. 

Moreover, an honest attempt was made to address all of the reviewer´s comments. They were very helpful and seemed fair to us. Also, all figures were checked and adapted with pace.com.

I am pleased about the cooperation and would still be very happy about the publication at PLOS. 

If any further questions arise, I am available at any time.

Kind regards from Witten!

Lukas Markert

---

## [Decision Letter · Decision Letter 1]

17 Feb 2021

Small RNAs as biomarkers to differentiate benign and malign prostate diseases: an alternative to transrectal punch biopsy of the prostate?

PONE-D-20-39305R1

Dear Dr. Markert,

We’re pleased to inform you that your manuscript has been judged scientifically suitable for publication and will be formally accepted for publication once it meets all outstanding technical requirements.

Kind regards,

Jean-Marc A Lobaccaro, PhD

Academic Editor

PLOS ONE

Additional Editor Comments (optional):

Reviewers' comments:

Reviewer's Responses to Questions

**Comments to the Author**

1. If the authors have adequately addressed your comments raised in a previous round of review and you feel that this manuscript is now acceptable for publication, you may indicate that here to bypass the “Comments to the Author” section, enter your conflict of interest statement in the “Confidential to Editor” section, and submit your "Accept" recommendation.

Reviewer #1: All comments have been addressed

2. Is the manuscript technically sound, and do the data support the conclusions?

Reviewer #1: (No Response)

3. Has the statistical analysis been performed appropriately and rigorously? 

Reviewer #1: (No Response)

4. Have the authors made all data underlying the findings in their manuscript fully available?

Reviewer #1: (No Response)

5. Is the manuscript presented in an intelligible fashion and written in standard English?

Reviewer #1: (No Response)

6. Review Comments to the Author

Reviewer #1: My requests are mostly addressed by the authors and I recommend accepting this manuscript in its current form

7. PLOS authors have the option to publish the peer review history of their article (what does this mean?). If published, this will include your full peer review and any attached files.

Reviewer #1: No

---

## [Editor Report · Acceptance letter]

24 Feb 2021

PONE-D-20-39305R1 

Small RNAs as biomarkers to differentiate benign and malign prostate diseases:an alternative for transrectal punch biopsy of the prostate? 

Dear Dr. Markert:

I'm pleased to inform you that your manuscript has been deemed suitable for publication in PLOS ONE. Congratulations! Your manuscript is now with our production department. 

Kind regards, 

on behalf of

Dr. Jean-Marc A Lobaccaro 

Academic Editor

PLOS ONE